# A Study on Energy Efficiency of a Distributed Processing Scheme for Image-Based Target Recognition for Internet of Multimedia Things

**Adel Soudani** [1,*], **Manal Alsabhan** [2] and **Manan Almusallam** [2]

1 Department of Computer Science, College of Computer and Information Sciences, King Saud University, Riyadh 11543, Saudi Arabia
2 Department of Computer Science, College of Computer and Information Sciences, Imam Mohammad Ibn Saud Islamic University (IMISU), Riyadh 11432, Saudi Arabia; mksabhan@imamu.edu.sa (M.A.); mmmusallam@imamu.edu.sa (M.A.)
* Correspondence: asoudani@ksu.edu.sa

**Abstract:** A growing number of services and applications are developed using multimedia sensing low-cost wireless devices, thus creating the Internet of Multimedia Things (IoMT). Nevertheless, energy efficiency and resource availability are two of the most challenging issues to overcome when developing image-based sensing applications. In depth, image-based sensing and transmission in IoMT significantly drain the sensor energy and overwhelm the network with redundant data. Event-based sensing schemes can be used to provide efficient data transmission and an extended network lifetime. This paper proposes a novel approach for distributed event-based sensing achieved by a cluster of processing nodes. The proposed scheme aims to balance the processing load across the nodes in the cluster. This study demonstrates the adequacy of distributed processing to extend the lifetime of the IoMT platform and compares the efficiency of Haar wavelet decomposition and general Fourier descriptors (GFDs) as a feature extraction module in a distributed features-based target recognition system. The results show that the distributed processing of the scheme based on the Haar wavelet transform of the image outperforms the scheme based on a general Fourier shape descriptor in recognition accuracy of the target as well as the energy consumption. In contrast to a GFD-based scheme, the recognition accuracy of a Haar-based scheme was increased by 26%, and the number of sensing cycles was increased from 40 to 70 cycles, which attests to the adequacy of the proposed distributed Haar-based processing scheme for deployment in IoMT devices.

**Keywords:** Internet of Multimedia Things (IoMT); WMSN; multimedia sensing; feature extraction; object recognition; low energy processing; Haar wavelet transformation; general Fourier descriptors (GFDs); distributed processing





## 1. Introduction

The Internet of Things (IoT), defined as the interconnection of individual embedded devices, is declared a suitable architecture for data collection in smart environments. To provide multimedia-based services and applications, the IoT expands to the Internet of Multimedia Things (IoMT) [1], a network that enables multimedia data exchange and collection. IoMT applications utilize wireless multimedia sensor networks (WMSNs) [2] to implement image-based object recognition and tracking that is otherwise difficult or impossible, especially in remote and high-risk environments such as the wilderness. The IoMT faces the same challenges as WMSNs and IoT in terms of energy efficiency, protocol design, and the need to optimize the network lifetime. In depth, multimedia communications require exchanging a significant amount of data, which increases the requirements of an IoMT-based platform in terms of bandwidth, memory, and computational resources. The major challenge in the deployment of these systems is to extend the network lifetime

for the availability of the provided service [3]. To address this problem, several research efforts proposed energy-efficient routing protocols [4], data compression algorithms [5], and distributed processing models [6]. Unprocessed image transmission to users could flood the network with intensive data bursts causing significant energy consumption, which can severely reduce the network lifetime. One of the potential solutions to this problem would be to process the captured image locally and to communicate only the useful data to the remote-control server through the network. While this approach looks very suitable for the image-based recognition application, it needs a careful effort to design a low-complexity sensing scheme that provides a tradeoff between the accuracy of target recognition and the energy-saving at the source sensor [7,8]. The accuracy of the target recognition in this approach depends basically on the efficiency of the extracted features to ensure a high discrimination level between different objects. The extraction of important features for the classification process requires applying image transforms that are often characterized by high complexity and might not be very adequate for sensor processing capability. We think that the idea of in-network cooperative execution of the event-based sensing scheme over a cluster of nodes can balance the processing task load over the network. Consequently, it will reduce per-node energy consumption in one sensing cycle, which is expected to significantly extend the network lifetime. This approach represents a potential efficient solution to the problem of image-based target recognition in IoMT. In this context, this paper presents a novel distributed energy-efficient scheme in which a cluster of nodes works collaboratively to identify targets in images. This scheme uses efficient shape descriptors to locally identify the target, rather than streaming captured images to the end-user to verify whether the captured image contains the event of interest. The execution of the scheme in a distributed model can provide energy-efficient performance compared to a centralized model in which a camera sensor depletes its batteries rapidly and undermines the practicality of the application.

Nevertheless, this event-based sensing scheme has substantial potential for efficiency if it can balance a moderate amount of computational complexity while ensuring a high level of accuracy in target recognition. The significant contribution of this paper is the development and implementation of a low-complexity distributed sensing scheme based on object feature extraction. The objective of this study is twofold; it aims at the first level to prove that the distributed processing of the scheme contributes to extending the network lifetime. At another level, the paper addresses the performance comparison of two methods for extracting relevant features, one using Haar wavelet transform and another based on general Fourier descriptors (GFDs) [9,10].

This paper describes and details experiments in which the performance of a distributed sensing system is evaluated. It analyzes the obtained result for the specified methodology, in which the capability of low-power sensing and notification is discussed. An important innovation of this scheme is its ability to reduce communication overhead and per-node energy consumption while ensuring that user notifications are delivered effectively. The obtained performances show that the proposed scheme outperforms similar schemes in recognizing targets and enables substantial energy saving.

## 2. Related Works

Over the last few years, IoT deployment has been presented as a promising solution for future data-based sensing architecture. Several research contributions are advancing the design of new paradigms to enable new services and applications based on these IoT devices [11,12]. Multimedia sensing based on IoT devices was the focus of research efforts to propose energy-efficient approaches that are intended to extend the network lifetime [1–3]. We can globally categorize them as follows:

### 2.1. Compression-Based Approaches

Among the most important solutions proposed to extend network lifetime are those based on data compression techniques in multimedia sensors to reduce the data transmitted to the end-user [13–15]. Despite the reduced bitrate achieved by this method, it was

demonstrated in most of these research works that the standard compression algorithms were designed for resourceful machines and therefore not adequate to be used in the context of WMSNs since they require high computational resources and consequently high processing power.

In [16], Leila et al. proposed a new method of low-complexity image compression based on a fast zonal DCT transform which allows a tradeoff between the quality of the reconstructed image at the receiver side and the energy required for communication. Even though the energy consumption was reduced in comparison to the JPEG standard, it remains high and does not encourage real deployment.

In [13], Kaddachi et al. proposed hardware implementation of compression 2D-DCT-based transform for image compression. The result of their implementation has shown a high energy gain; however, the implementation cost and the low scalability of this solution limit its adaptability to WMSNs.

Wang et al. [17] proposed a multiresolution compression and query (MRCQ) technique for in-network data compression. The given method was used to reduce the amount of data to be transmitted to the end-user. The proposed method was evaluated and has shown its efficiency for scalar data sensing. However, it was not applied for image sensing where the compression needs much higher computation.

This technique of multiresolution was also used in [18] to manage data sorting and query processing. However, this technique was not applied to multimedia content.

Another method for ROI extraction and compression was proposed in [19] based on discrete Tchebichef transform (DTT). The results have shown interesting energy reduction of the generated data to be transmitted when compared to the application of standards such as JPEG and the binary DCT-based compression technique. While this method looks interesting, the activity of the wireless transceiver will be proportional to the size of the extracted ROI. Furthermore, the proposed approach will not avoid the transmission of unnecessary data to the end-user since it is not based on in-network event recognition.

### 2.2. Distributed Processing Approach

Alternatively, distributed compression was also proposed as a solution to reduce per-node energy consumption. In depth, the distribution of the computation over several nodes allows for balancing the processing load, saving node energy, and reducing the size of transmitted data [3]. A distributed image compression algorithm based on wavelet transformation was presented and evaluated by Wu et al. in [20,21]. The authors proposed a distributed compression scheme where nodes compress the image during its forwarding to the destination under some quality constraint; the results of the simulation have shown an extension in the network's lifetime and demonstrate the efficiency of collective processing of the image. However, their approach aims only to reduce the amount of data without in-network recognition of a specific event which might considerably reduce the energy consumption.

In [22], the authors evaluate the performance of a cluster-based hybrid computing paradigm for collaborative sensing and processing in WSNs versus distributed computing paradigms, such as the mobile agent model and client/server model. Based on the results, the model was found to be energy-efficient and, therefore, scalable. However, the effectiveness of heavy data processing, such as images or videos in WMSNs, requires further investigation.

In [23], Qi et al. proposed a distributed multisensory method for detecting targets. In this approach, the target is detected from different angles when it enters the detection area boundaries. Following this, the node will aggregate a notification and send it to the base station to announce the location of the moving target. Based on the performance analysis, centralized processing using a single node is less likely to improve detection probabilities than collaborative node sensing.

Lin et al. [24] presented a distributed approach to recognizing a given identity by extracting features from images. In the first step, the scheme extracts and detects the face region, and then the components of the face are detected. Parallel processing is used to distribute the components among nodes for processing. Nevertheless, this work has some

limitations in computing and processing sharing, and it would be helpful to determine the algorithm's reliability across different network scales.

A clustering approach is employed in [25] to detect and track an identified target. In this work, however, acoustic and visual sensors are combined with passive infrared motion detectors. Additionally, object identification is accomplished at the sink node. In contrast, ref. [26] used audio-visual and scalar features to enhance the recognition and classification of objects. As a result, network traffic was significantly reduced, which increased the network's lifespan.

Based on the image fusion algorithms, Latreche et al. [6] produced a final informative image using a combination of images captured from several different angles and distances within the monitoring area. The integer lifting wavelet transforms (ILWTs) and the discrete cosine transform (DCT) were used to produce high-quality blended images using this hybrid multimedia image fusion. Two steps were then performed on the fused image: extracting low-frequency coefficients from the image, followed by a second step to capture satisfactory detail coefficients from the same image. Despite the demonstrated detection accuracy of this approach, it is still necessary to evaluate the energy efficiency of ATmega128 microcontrollers.

Camera-equipped nodes serve as cluster heads in a distributed two-hop clustered image transmission scheme [27], distributing compression tasks among the cluster members. These nodes also participate in the distributed compression and transmission processes. By balancing the energy consumption between the cooperating nodes, the operation of the network can be extended. Despite this, the transmission of a continuous stream of images drains the network's energy and increases contention and congestion. The authors used a hardware platform for energy conservation in [28], but the platform was not considered a scalable solution because of the estimated high cost of implementation.

The work presented in [29] introduces an energy-aware collaborative tracking and moving detection scheme for WMSNs. The proposed method relies on collaboration among sensors to extract a lightweight image from a multiply captured scene to reduce computation and communication costs.

*2.3. Event-Based Detection Approach*

Using a local event-based sensing and detection scheme would be an effective method for maximizing energy efficiency and extending the network lifetime. Using this technique reduces redundant image data and network traffic [7,30,31]. This may be achieved by using a regional interest descriptor locally at the network level to determine if the image contains an even area of interest and send the minimum data. By using this approach, the amount of data transmitted to the sink node can be reduced. This preserves the energy of the source sensor and the energy of the other nodes of the network. The implementation of this technique requires the creation of a robust image analysis method independent of translation, orientation, and scaling properties.

An image of an object captured by a WMSN is divided into small blocks using a motion detection framework [32]. The differences between the blocks of the captured image and those of the reference image are then identified. In addition to reducing energy consumption, this approach also reduces bandwidth consumption. The present study is intended for applications involving the detection of objects' appearances, in which the identification and classification processes are shifted to the base station. Using the Haar wavelet implemented locally in the sensor, Vasuhi et al. [33] have extracted object features from WMSNs, but they have not considered the scheme's computational complexity or power consumption.

Another method for pattern recognition is based on an artificial immune system [34], but it had a high associated energy consumption; therefore, it is not suitable for sensors with limited energy resources. In [30], the author used a shape-based descriptor to identify targets at the source node. The results obtained indicate a significant reduction in power consumption. Unfortunately, the centroid distance and the curvature signature used for the recognition capability of the presented scheme suffer from inaccuracies. In particular, the proposed scheme has high variance and sensitivity to the properties of the detected

objects in the images. Additionally, the proposed scheme was based on the assumption that only one object appeared in the camera's field of view.

Many other solutions were also proposed to save energy that will not be detailed here, such as those found in [32,35,36]. They addressed the idea of extending the network lifetime from different approaches.

Among these reviewed approaches, we consider the idea of distributed processing as a potential solution to extend the whole network lifetime by reducing per-node energy consumption. The approach of a distributed image-based target recognition sensing scheme was not yet studied. We propose a new scheme for image-based target recognition that will be processed over a cluster of nodes dynamically created for this purpose. We focus on the design and the performance evaluation to demonstrate the energy efficiency of this scheme.

## 3. Methodology

This section will present the specification of our distributed event-based object detection and recognition scheme. This scheme is expected to reduce the energy consumption per node and extend the network lifespan. We will illustrate the required pipeline steps toward making the recognition decision. At the end of this section, we will demonstrate the design of the distributed cluster through the network and energy consumption models. In surveillance applications, a camera node regularly senses the surrounding neighborhood, thus creating a stream of visual data. In the approach presented in this paper, instead of blindly compressing any captured image and possibly flooding the network with irrelevant data, the sensor utilizes a low-complexity feature extraction method to detect the event of interest. This way, the end-user will only be notified when an event of interest is detected.

As an initial step, the end application configures the processing sensors by setting up a target signature, distinguishing features used to recognize a specific target. Once an event is sensed, the camera starts the sensor clustering and performs the distributed object detection and extraction process from the captured scene. In depth, the features extracted from the identified object are compared against the target signature. If a substantial similarity between the signatures is detected, a notification is sent to the end-user. Otherwise, the sensed event is rejected, and the camera sensor restarts the event-based sensing. This study aims to demonstrate an energy-efficient and scalable scheme for target detection by minimizing computational complexity, increasing detection accuracy, and reducing per-node storage requirements and communication overhead.

### 3.1. Local Event Detection

In this work, the sensing cluster has to detect the appearance of a new object and locally decide if the detected object is of interest to the application, and accordingly, the network might be involved in data transmission. The different steps that represent the proposed scheme are as follows:

#### 3.1.1. Background Subtraction

Background subtraction is commonly used in WMSN applications to separate the object's region of interest (ROI) by calculating the difference between the pixel intensity in the captured scene frame (foreground image) and the pixel intensity of a static scene frame (background image).

Many background subtraction methods presented in the literature vary in computational complexity, storage requirements, and detection precision [37]. A simple background subtraction with a low computational cost uses the running Gaussian average. This approach maintains rapid isolation, high accuracy, and low memory occupancies [38,39].

Assuming a grayscale image ($M$) composed of $p \times p$ pixels, the background pixel value at frame n is updated by running a Gaussian probability density function as follows:

$$\beta_n = \beta_{n-1} + \alpha(F_n - \beta_{n-1}) \tag{1}$$

where $\beta_n$ is the updated background average, $F_n$ is the current frame intensity, $\beta_{n-1}$ is the previous background average, and $\alpha$ is an updating constant whose value ranges between 0 and 1 and represents a tradeoff between stability and quick update.

A pixel is classified as foreground (i.e., belongs to an updated object) if the condition expressed in (2) is met:

$$\hat{M} = \begin{cases} 1 \ \ if \ |F_n - \beta_{n-1}| > Thr_{Background} \\ \ \ 0 \ \ \ \ \ Otherwise \end{cases} \tag{2}$$

where $\hat{M}$ is a binary image and $Thr_{Background}$ is the threshold for background subtraction.

### 3.1.2. The ROI Extraction

To identify and recognize an object of interest, we need to extract the set of blocks found in the 2D pattern $\hat{M}$ that represents the ROI. In the literature, various methods are used to distinguish the ROI from the captured scene, such as row and column scanning functions [38], an iterative threshold approach [39], and a region operating segmentation algorithm [40]. However, many of these separation algorithms are characterized by high computational complexity and are appropriate for computers with high processor capabilities, such as those used in vision applications [41].

In our research, we focus on a tradeoff between processing complexity and detection accuracy. Therefore, we adopt an iterative thresholding algorithm. This approach is based on a predetermined threshold value for extracting the region of interest (ROI). The algorithm subdivides the image into sub-blocks and counts the total number of pixels participating in the appeared object. Assuming an object block denoted by $\beta_n(j)$ and a background block denoted by $\beta_{n-1}(j)\beta_{n-1}(j)$, a new object is detected when the difference between the image blocks is significantly higher than a certain threshold $Thr_{Region}$, as expressed in the following:

$$\beta_n = \beta_{n-1} + \alpha(F_n - \beta_{n-1}) \tag{3}$$

This approach significantly reduces the memory requirements and improves the energy consumption level related to pixel processing compared to row and column scanning and region-growing algorithms.

### 3.1.3. Extraction of Features' Vectors

1. Haar Wavelet Transformation

Our presented work is motivated by the need for an energy-efficient pattern recognition approach for target matching. The discrete wavelet transformations (DWTs) are mentioned in the literature for application tracking applications that extract the object features in different scales and sub-bands [42]. The Haar transformation technique is the simplest form of wavelet transformation, where the main advantage over the Fourier transformation is its ability to capture both the frequency and location information [43,44]. The Haar wavelet transformation is a promising low-complexity reversible feature extraction method that requires low memory occupancy.

In general, wavelet decomposition transforms an image into approximation and detail coefficients [45]. The approximation coefficients show the general trend of pixel values, while detail coefficients show the details and changes in an image. Assuming a 2D image $f(x, y)$, with x rows and y columns, a two-dimensional Haar transform first performs a 1D row-wise transform to produce an intermediate result $f'(x, y)$. It then performs a 1D column-wise transform on $f'(x, y)$ to produce the final result $f''(x, y)$ that consists of four different sub-bands: $f_{LL}(x, y)$, $f_{LH}(x, y)$, $f_{HL}(x, y)$, and $f_{HH}(x, y)$, where $f_{LL}(x, y)$ represents the low-pass (average or smooth) coefficients, and $f_{LH}(x, y)$, $f_{HL}(x, y)$, and $f_{HH}(x, y)$ represent the high-pass (detail) coefficients. Further image decomposition uses $f_{LL}(x, y)$ as an input image as depicted in Figure 1.

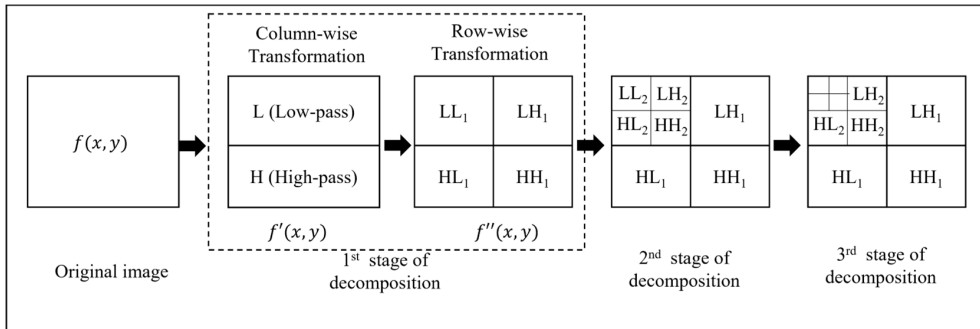

**Figure 1.** Computation of 2D Haar wavelet transformation.

The approximate and detailed components of the Haar transform [10] are given by

$$L[n] = \frac{input[2n] + input[2n+1]}{2} \tag{4}$$

$$H[n] = \frac{input[2n] - input[2n+1]}{2} \tag{5}$$

where *input* is either a 1D row or column and *n* is the pixel location within that row/column. Equation (4) computes the average of adjacent pixel pairs, a process referred to as low-pass filtering. On the other hand, Equation (5) computes the difference between the same pair of pixels, a process referred to as high-pass filtering.

The Haar wavelet transformation is not invariant to scale, translation, and rotation object properties [43]. Therefore, a pre-processing normalization phase should be considered before the matching process. The translation invariance property is achieved by using regular moments to calculate the centroid location of the original image [46]. The two-dimensional geometric regular moment of order (p + q) of a function f(x,y) is defined as follows:

$$m_{pq} = \sum_{x=1}^{M} \sum_{y=1}^{N} x^p y^q f(x, y) \tag{6}$$

where p, q = 1, 2 ... ∞.

Equation (6) can be used to generate three moments: m00, m01, and m10, which are then plugged into the following equations to move the region of interest to the center of the image:

$$\acute{x} = \frac{m10}{m00} \ and \ \acute{y} = \frac{m01}{m00} \tag{7}$$

$$g(x, y) = f\left(\frac{x + \acute{x}}{a}, \frac{y + \acute{y}}{a}\right) \tag{8}$$

On the other hand, scale invariance is achieved by either maximizing or minimizing the target such that the zeroth-order moment ($m_{00}$) is equal to predetermined idle values. The scale (*a*) is a scale factor that can be computed as follows:

$$a = \sqrt{\frac{\beta}{m00}} \tag{9}$$

where ($m_{00}$) defines the total number of pixels in a reference image and β is the objective total number of pixels.

The use of wavelet-based methods for pattern matching and recognition is either constrained to objects in fixed rotation alignments or involved in an extensive training set of objects in all possible orientations. The ring projection transformation is used to transform the extracted 2D Haar wavelet coefficients into a 1D signal as a function of radius to reduce the computational cost and make the matching invariant to rotation [43,47].

Let the pattern of interest be contained in a circular window of radius ($w$) where the image size is not less than $20 \times 20$ pixels according to the recommendation in [43]; the ring projection of the wavelet coefficients is calculated as follows:

$$P(r) = \frac{1}{n_r} \sum_k f_d(r \cos\theta_k, r \sin\theta_k) \qquad (10)$$

where $f_d$ is the transformed 2D coefficients into polar coordinates, and $n_r$ is the total number of pixels falling on circle radius $r$, $r = 0, 1, \ldots w$.

2. General Fourier Transformation

The general Fourier descriptor (GFD) [9,48] is a mathematical model that uses Fourier transformation to transform a shape signature into a set of descriptor features. First, GFD transforms the input image f($x_i$,$y_i$) of size N × M where f is defined by {f($x_i$,$y_i$): 1≤ i ≤ M, 1 ≤ j ≤ N} into a polar image f(r,θ) using the following equations:

$$r = \sqrt{\left((x - \acute{x})^2 + (y - \acute{y})^2\right)} \qquad (11)$$

$$\theta = \tan^{-1}(y - \acute{y}/x - \acute{x}) \qquad (12)$$

where $\acute{x}$ and $\acute{y}$ are the mass center of the shape. Then, the Fourier transformation takes place to extract the signature feature vector set, referred to as Fourier descriptors (FDs), using the following equation:

$$FD(\rho, \varphi) = \sum_{r=0}^{R} \sum_{\theta_i=0}^{T} f(r, \theta_i) e^{j2\pi(\frac{r}{R}\rho + \frac{2\pi i}{T}\varphi)} \qquad (13)$$

The parameters $\rho$ and $\varphi$ reflect the image size, $\theta_i = \frac{i2\pi}{T}$, $0 \leq \varphi < T$; R is radial resolution, and T is angular resolution.

The GFD method is invariant to translation. However, to achieve rotation and scaling invariance, a normalization step is applied to the extracted feature vector set, as in the following:

$$GFD = \left\{ \frac{|FD(0,0)|}{area}, \ldots, \frac{|FD(0,n)|}{|FD(0,0)|}, \ldots, \frac{|FD(m,n)|}{|FD(0,0)|} \right\} \qquad (14)$$

where m is the maximum number of radius frequencies and n is the maximum number of angular frequencies. Zhang et al. indicate the efficient shape descriptors using GFDs for shape representation are 52 feature descriptors where radial frequencies $m = 4$ and angular frequencies n = 9 [49]. We refer to the GFDs collectively as the detected signature $\widetilde{S}$.

3.1.4. Target Recognition

There are various ranking measurement functions presented in the literature [9,30,50], which can be employed to assess the comparison between the matched signature and identified object. However, due to the sensor design limitations, there is a need to adopt a minimal computational complexity measurement to avoid the processing overhead of local target detection.

For the matching process, we use a similarity function $\Delta$ to measure the distance between the extracted object signature ($\widetilde{S}$) and a reference signature ($S$). The similarity function is correlated with a threshold ($T$) that signifies the level of similarity between the compared signatures. If the measured distance is less than the threshold ($T$), the detected object is declared as a target, and the sensor notifies the end-user; otherwise, the detected object is ignored. Section 4 provides more details on the experimental dataset used in threshold evaluation.

We use the Euclidian distance (ED), due to its lightweight computational complexity, as a matching statistical measurement to compare the extracted object signature ($\widetilde{S}$) against a reference signature ($S$) as in (15):

$$ED = \left( \sum_{i=1}^{N} \left( X_i - \overline{X}_i \right)^2 \right)^{0.5} \tag{15}$$

where $N$ represents the total vector set, $X_i$ denotes the $i$th feature vector of the extracted signature, and $\overline{X}_i$ denotes the $i$th feature vector of the reference.

### 3.1.5. End-User Notification

For significant energy consumption reduction and time saving, the remote end-user application can configure the notification preference to one of three options:

- A one-byte message representing the recognition status, i.e., yes/no message;
- The extracted feature set that represents the recognized object;
- The extracted region of interest itself.

This on-demand notification configuration will minimize bandwidth congestion by reducing the data size and the load of communication traffic, which positively impacts the network lifetime.

The proposed scheme for image-based target recognition has the following algorithmic complexities summarized in Table 1.

**Table 1.** Computational complexities of the two proposed schemes.

| Tasks | Computational Complexity |
|---|---|
| Background subtraction | O (N.N) |
| ROI extraction | O (N.N) |
| Feature extraction based on 2D Haar wavelet transform | O (N.N) |
| GFD feature extraction based on 2D Fourier transform | O (N.N.N) |
| Matching using ED | O (N) |

Based on Table 1 we can note that the complexity of the scheme based on the GFD descriptor has higher complexity than the scheme based on Haar wavelet transform O (N.N). This result promises that the scheme based on Haar wavelet transform will be less demanding in terms of computing resources and energy consumption when deployed in IoT devices.

### 3.2. Distributed Processing Cluster Design

For multimedia sensing applications, preserving the in-node energy, which extends the network lifetime, is one of the leading design challenges [21–23,51]. One of the attractive energy-efficient solutions is to use cluster-based processing that preserves the resident energy by distributing the implementation using a collective network synergy to achieve better performance compared to the centralized implementation. In this section, we will detail the adopted network model and the energy consumption model.

### 3.2.1. Network Model

This work is by inspired the processing cluster design from the LEACH-C protocol [52] and adapts it to distributed processing requirements. Our goal is to build a scalable and energy-efficient distributed processing cluster for image-based target recognition. In this paper, we assume the following specification in the network design:

- The network consists of constant wireless camera sensors in each processing cluster and regular static sensors used for processing and communication assignments.
- The camera sensors are scattered in predefined locations that consent to target detection.
- The density of the network nodes must be high and randomly distributed for complete area coverage.
- The camera node will lead the processing cluster initiation and termination. The selection of cooperating nodes mainly depends on the maximum resident energy level.
- The network will be deployed in a low-dynamic environment, such as the natural habitat of wild animals.

The event-based sensing scheme is an iterative process where the camera begins the sensing cycle by capturing an object in the surveillance scene. Then the camera starts

initiating the distributed processing cluster formation by selecting the nodes with the highest resident energy levels. After completing the cluster-forming process, the camera distributes the processing tasks across the cluster cooperating nodes to accomplish target identification and recognition. The scenario of a single processing cycle, depicted in Figure 2, is described as follows:

- The camera node broadcasts the [ENERGY_REQUEST] packet to neighbors' nodes towards creating a processing cluster.
- After, the camera's neighborhood processing nodes will respond with their residual energy using the [ENERGY_RESPONSE] packet. Then, the camera retains the candidates' list of responding processing nodes ranked by their energy level and selects two candidate nodes of the processing cluster P1 and P2, based on the highest resident energy level.
- The camera node uses the [JOIN] packet to delegate a particular task for each participating cooperating node of the cluster.
- The selected cooperating nodes P1 and P2 should approve their participation by sending [FORM] acknowledgment packets. Otherwise, go back to step 1.
- After forming processing clusters, the camera will apply the object extraction method to the detected object and send it to the first node for further processing through the [ROI] packet.
- Then, P1 and P2 will work together on object feature extracting, matching, and notifying steps, where P1 is responsible for feature extraction on the functional ROI obtained from the camera, and P2 will accomplish the matching and notification step. Once the object is detected, the P2 responsible for the matching process will notify the camera.
- At the end of the processing cycle, the camera will notify the end-user when the extracted signature matches a reference signature. Otherwise, the detected object is discarded, and finally, the camera will terminate the current sensing cycle and start a new cycle again.

After each sensing cycle, the notification will be sent based on the event of interest. The cluster will be terminated. The cooperating nodes will be available for any other network processing activity unless the camera chooses them again in the following sensing cycle. This work uses the packet payload segment of the IEEE 802.15.4 standard to set the control fields needed to establish and process the camera cluster. Our algorithm's contribution in designing the communication packet is to rely on the payload field without modifying the standard IEEE 802.15.4 packet header structure. This advantage gives the scheme the flexibility to design ten different control messages and four data exchange messages using the structure of payload fields, as illustrated in Figure 3.

We based communication and processing packets on the following communication control requests to be exchanged between the nodes to ensure the setup of the processing cluster. They are used for cluster establishing and processing by splitting the algorithm tasks between the cooperative nodes.

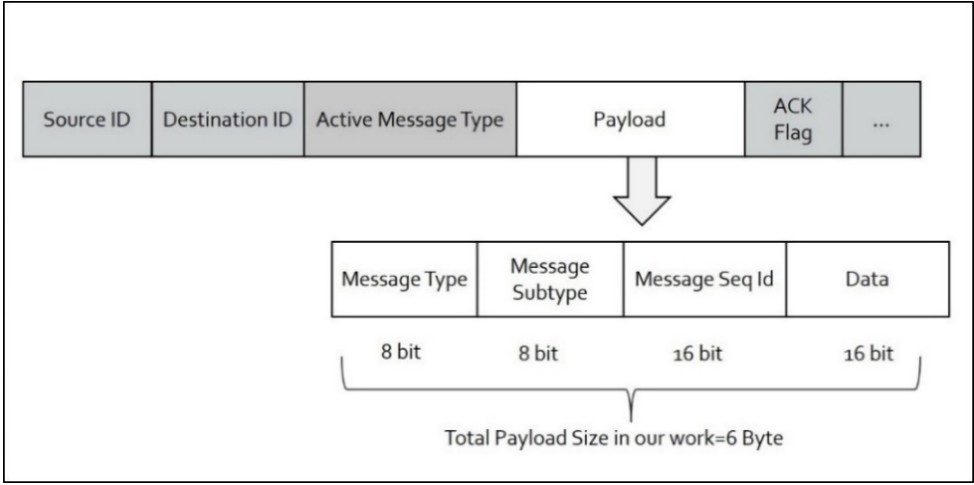

**Figure 2.** The scenario of the proposed distributed processing scheme.

**Figure 3.** Packet structure used in the scheme.

### 3.2.2. Energy Consumption Model

In this work, we use the following energy consumption model used in LEACH [53] as illustrated in the following equations:

$$E_{tx}(l,d) = \begin{cases} l \times E_{elec} + l \times E_{mp} \times d^4, & if\ d \geq d_0 \\ l \times E_{elec} + l \times E_{fs} \times d^2, & if\ d < d_0 \end{cases} \tag{16}$$

where $E_{elec}$ is the energy consumed by the circuit per bit; $d$ is the distance between sender and receiver; $E_{fs}$ relates to free space energy depleted by the amplifier for a short distance, while $E_{mp}$ relates to multipath fading energy that is drained by the amplifier and long distances. $d_0 = \sqrt{E_{fs}/E_{mp}}$ is the reference distance between the sender and receiver. If this distance is less than $d_0$, then $E_{fs}$ is turned on; otherwise, $E_{mp}$ is demanded.

The energy consumption of a sensor node when it receives a *k*-bit packet is as follows:

$$E_{rx} = k \times E_{elec} + k \times n \times E_{DA} \tag{17}$$

where $E_{DA}$ is the energy needed to aggregate data, $k$ is the number of bits per packet, and $n$ is the number of received messages.

## 4. Result and Discussion

In this section, we evaluate the accuracy of the proposed scheme in terms of object recognition, and we estimate its energy consumption when implemented on constrained sensors.

### 4.1. Experiment Setup and Parameters

We assume a network area of $100 \times 100$ m. The camera node is in the center of the area at position (50,50). The sink node is located at position (0,0). N sensor nodes are scattered in random positions. They represent the set of nodes in the coverage of the camera node antenna. For this experiment, we assume N = 10. Table 2 lists the sensor specification used in this simulation. To evaluate the energy consumption level associated with the internal algorithm processing, we used the Avrora simulator [54]. Matlab software (MathWorks, Natick, MA, USA) was used to simulate the communication between the sensor node and the sink and assess the algorithm's recognition capability.

**Table 2.** Sensor specifications.

| Criteria | Description |
| --- | --- |
| Mote Series | Mica2 |
| Sensor Processor | ATmega128L, 868/916 MHz |
| Measurement Flash | 512 K Bytes |
| Program Flash Memory | 128 K Bytes |
| Sensor Data Rate | 38.4 Kbaud = 20/40 Kbps |
| Network Communication Model | Based on Signal Strength |
| Initial Energy | 100 mJ |
| Electric Consumption Energy (RX, TX) | $5 \times 10^{-5}$ mJ/bit |
| Transmit Amplifier Efs | $1 \times 10^{-8}$ mJ/bit/m$^2$ |
| Transmit Amplifier EMP | $1.3 \times 10^{-12}$ mJ/bit/m$^4$ |
| Data Aggregation Energy (EDA) | $5 \times 10^{-9}$ mJ/bit/signal |
| Sqrt (Efs/Emp) d0 | 8.7705 m |

### 4.2. Target Recognition and Performance Analysis

The proposed scheme combines the Haar wavelet decomposition with transformation of ring projection (TRP) [43] to extract object features. An object is recognized when the distance between the extracted features and the features of a reference image crosses a predefined threshold. To demonstrate the algorithm's capabilities, we tested its performance using images of size $64 \times 64$ pixels 8 bpp and $128 \times 128$ pixels 8 bpp. We also developed a dataset of 168 images corresponding to six animals (horse, wolf, deer, ele-

phant, rhino, and tiger). The standard MPEG-7 dataset [55] used in the literature does not simulate different motor modes created by moving objects captured by an object-tracking application. Therefore, we generated 28 8-bitmap grayscale images for each class of animal. Each image manifests the animal in different orientations. This aim is achieved by placing the object in different positions and applying different levels of scaling and rotation (Appendix A). The image dataset was divided: 60% of the images were used to train the algorithm and learn threshold values, and 25% were used to evaluate the algorithm's accuracy. The remaining 15% of images were utilized as reference images.

Every image in the dataset undergoes a feature extraction process as follows: (1) apply the Haar wavelet transformation for object feature extraction to extract 2D approximate co-efficient vectors; (2) apply transformation of ring projection (TRP) to convert features into a 1D feature vector that is invariant to object rotation. The combined Haar-TPR feature extraction yields only 12 feature vectors to represent the extracted object. This minimum number of features implies lower memory requirements and reduced computational complexity of the signature-matching algorithm.

Figure 4 shows some results obtained from applying the feature extraction based on the Haar wavelet and using the GFD to all images related to a specific class of animal across different orientations. The colored curves represent the cumulative $i$th feature vector of a specific image in the class dataset. Comparing Figure 4a,b, the Haar-based extracted feature set using the horse class shows almost identical curves with a minor variation in the difference between them more than in the GFD-based algorithm. As demonstrated in the remaining graphs in the same figure, we can observe the high correlation level of the extracted features despite the differences in the object orientation using the Haar-based algorithm rather than the GFD technique. This attests to the Haar-wavelet-based features' stability in representing the animal and their invariance to levels of rotation, scaling, and translation applied to the object. Moreover, as we noticed, the total number of object descriptors is reduced from 52 feature vectors using the GFD-based algorithm to 12 feature vectors using the Haar-based algorithm. This improvement in reducing the needed number of descriptors can free up needed memory space and minimize the transformed data size.

As described earlier, the extracted features were compared against a pre-loaded target signature in the configuration setup. If the difference between the detected signature $(\widetilde{S})$ and the reference signature $(S)$ is less than a threshold $(T)$, the detected object is declared a target, and the sensor notifies the user. Otherwise, the detected object is ignored.

We adopted a minimum Euclidean distance (MED) metric [56] to find the possible discrimination threshold set value. We have to find the minimum Euclidean distance as follows:

$$MED = Min\left(ED_{Other\ Classes}\right) - Max(ED_{native\ classes}) \tag{18}$$

We repeatedly applied Equation (18) for each native class in the image dataset where we calculated the minimum and maximum ED for each training set in the native class and each training set in other classes, as shown in Figure 5. After we apply this process to all, we obtain a set of possible threshold values.

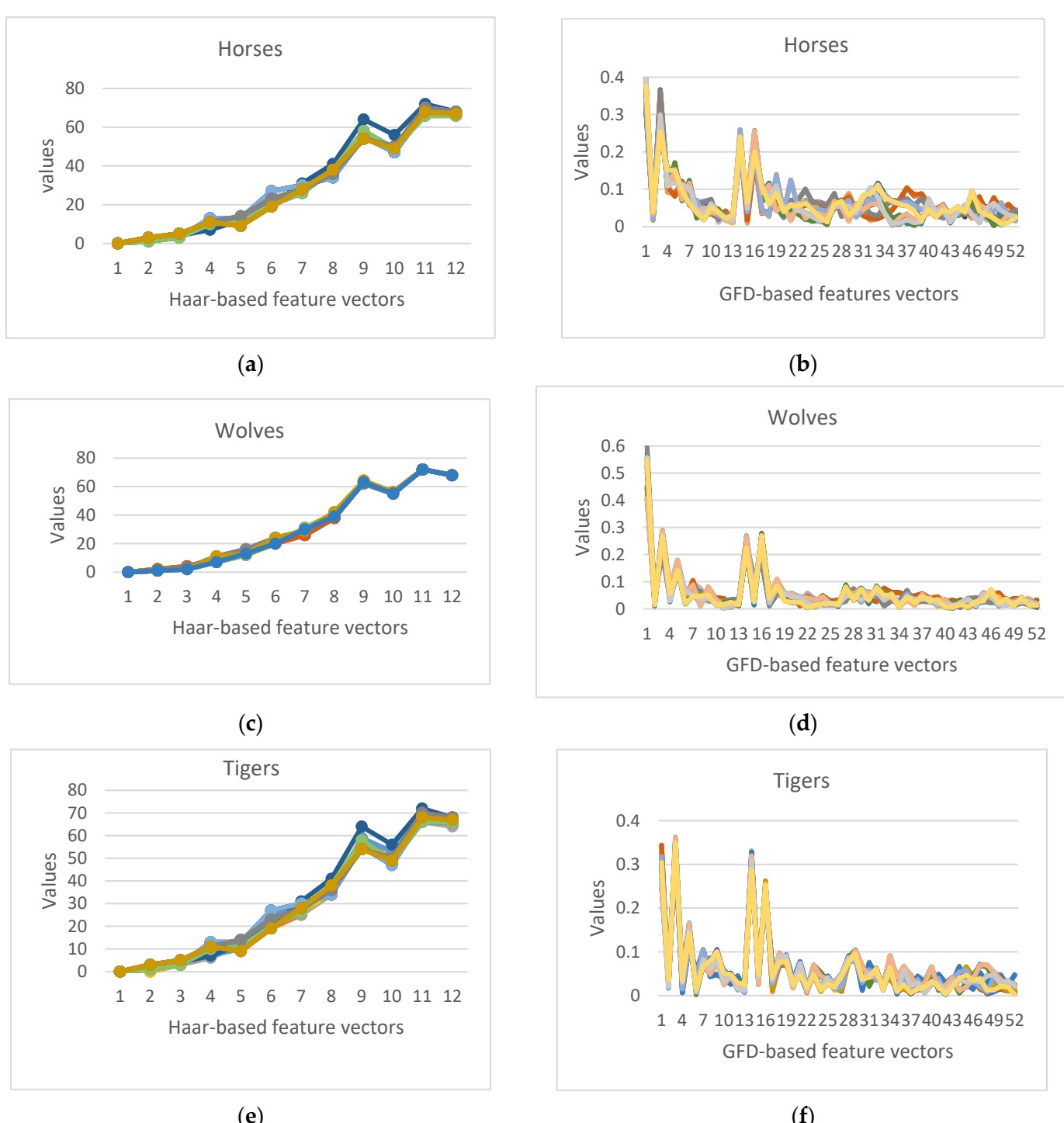

**Figure 4.** Haar-based and GFD-based feature extraction applied to some animal classes from the dataset; (**a**,**b**) Horses database, (**c**,**d**) Wolves dataset, (**e**,**f**) Tiger dataset.

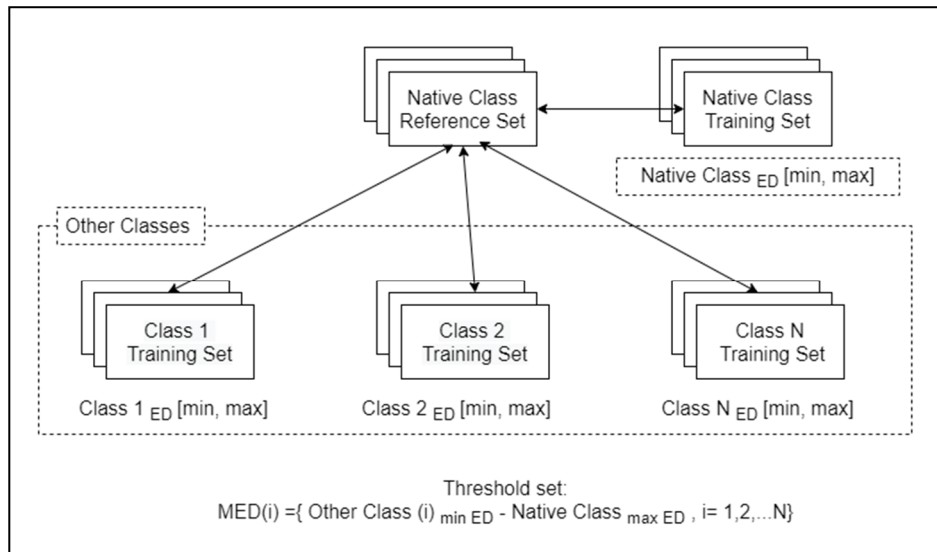

**Figure 5.** Extracting possible threshold values from a dataset.

We conducted six rounds of experiments to determine possible threshold values that optimize object recognition for all six classes of images. The performance of the object recognition algorithm is evaluated by measuring the retrieval performance using precision and recall metrics as illustrated in Equations (19) and (20), where TP is the total number of actual positive, relevant classified images, FP is the number of false positive classified irrelevant images, and N is the total number of relevant shapes in the dataset.

$$\text{Precison} = \frac{\text{TP}}{\text{TP} + \text{FP}} * 100 \tag{19}$$

$$\text{Recall} = \frac{\text{TP}}{\text{N}} * 100 \tag{20}$$

Classification and retrieval metrics at each threshold value are illustrated in Table 3. Note that setting the threshold to 7 achieves excellent classification efficiency across all animal classes. However, as we decrease the threshold value, we note a decrease in classification efficiency. The lowest classification efficiency of 92% was scored at a threshold value of 0 for the wolf animal class. The precision metric was stable at 100% across all animal classes for threshold values ranging from 0 to 5.

**Table 3.** Evaluation of Haar-based recognition scheme.

| Threshold Value | Classification Performance | Horse | | Wolf | | Deer | | Elephant | | Rhino | | Tiger | |
|---|---|---|---|---|---|---|---|---|---|---|---|---|---|
| | | Train | Test | Train | Test | Train | Test | Train | Test | Train | Test | Train | Test |
| 7 | Precision % | 94.4 | 94.4 | 100 | 100 | 100 | 100 | 100 | 100 | 100 | 100 | 95.7 | 97.7 |
| | Recall % | 100 | 100 | 100 | 100 | 100 | 100 | 100 | 100 | 100 | 100 | 100 | 100 |
| 5 | Precision % | 100 | 100 | 100 | 100 | 100 | 100 | 100 | 100 | 100 | 100 | 100 | 100 |
| | Recall % | 94.1 | 100 | 100 | 100 | 100 | 100 | 100 | 100 | 100 | 100 | 100 | 100 |
| 3 | Precision % | 100 | 100 | 100 | 100 | 100 | 100 | 100 | 100 | 100 | 100 | 100 | 100 |
| | Recall % | 94.1 | 100 | 100 | 100 | 100 | 100 | 100 | 100 | 100 | 100 | 100 | 100 |
| 2 | Precision % | 100 | 100 | 100 | 100 | 100 | 100 | 100 | 100 | 100 | 100 | 100 | 100 |
| | Recall % | 94.1 | 95 | 97 | 97 | 100 | 100 | 95 | 95 | 100 | 100 | 95.5 | 97.5 |
| 0 | Precision % | 100 | 100 | 100 | 100 | 100 | 100 | 100 | 100 | 100 | 100 | 100 | 100 |
| | Recall % | 94.1 | 95 | 94.6 | 94.6 | 100 | 100 | 94.1 | 94.1 | 100 | 100 | 95.5 | 97.5 |

Nevertheless, as we increase the threshold to 7, we note a precision as low as 94% regarding the horse class. The recall metric achieves a perfect score at threshold 7. However, as we decrease the threshold, the recall metric can become as low as 92% when evaluated at a threshold value of 0 for the wolf class. We can conclude that choosing a higher threshold will improve both classification and recall efficiency, but the retrieval precision will decrease gradually. Therefore, we select the middle thresholds 3 and 5 to test object recognition accuracy when applied to the testing dataset.

According to this result, our proposed scheme presents a robust and accurate shape descriptor for recognizing and identifying an object. It presents a high ability to capture significant features of the sensed object compared to obtained results from the GFD-based recognition scheme. However, as shown in Table 4, we can notice that the precision level varied within a short range of ED threshold values from 0.155 to 0.26 related to the feature vectors extracted using GFD, resulting in difficulty in classification efficiency in determining whether the target belongs to the specific class or not. In contrast, this ED range is expanded using the Haar-based scheme as shown in Table 3 and improves the classification in precision and recall metrics.

**Table 4.** Evaluation of GFD-based recognition scheme.

| Threshold Value | Classification Performance | Horse | | Wolf | | Deer | | Elephant | | Rhino | | Tiger | |
|---|---|---|---|---|---|---|---|---|---|---|---|---|---|
| | | Train | Test | Train | Test | Train | Test | Train | Test | Train | Test | Train | Test |
| 0.26 | Precision % | 46.8 | 50 | 34.5 | 48 | 100 | 100 | 53.5 | 60 | 33 | 47 | 46.2 | 50 |
| | Recall % | 100 | 100 | 100 | 100 | 100 | 100 | 100 | 100 | 100 | 100 | 100 | 100 |
| 0.19 | Precision % | 88.3 | 90 | 82.9 | 90 | 100 | 100 | 97.1 | 100 | 73.9 | 80 | 97.1 | 100 |
| | Recall % | 100 | 100 | 100 | 100 | 100 | 100 | 100 | 100 | 100 | 100 | 100 | 100 |
| 0.17 | Precision % | 94.9 | 97 | 98.5 | 100 | 100 | 100 | 100 | 100 | 95.7 | 98 | 100 | 100 |
| | Recall % | 94.1 | 95 | 100 | 100 | 95.5 | 98 | 100 | 100 | 100 | 100 | 100 | 100 |
| 0.165 | Precision % | 94.9 | 98 | 100 | 100 | 100 | 100 | 100 | 100 | 95.7 | 98 | 100 | 100 |
| | Recall % | 94.1 | 95 | 100 | 100 | 95.5 | 97 | 100 | 100 | 100 | 100 | 95.5 | 97 |
| 0.155 | Precision % | 100 | 100 | 100 | 100 | 100 | 100 | 100 | 100 | 100 | 100 | 100 | 100 |
| | Recall % | 94.1 | 95 | 100 | 100 | 95.5 | 97 | 100 | 100 | 100 | 100 | 95.5 | 97 |

For example, taking the horse class, using the highest threshold value will show a high ability of image retrieval with 100%, while the precision shows a low accurate classification ability in the GFD-based algorithm. Using the highest possible threshold value for the same class will maintain at least 94.1% classification accuracy. For the same class, the precision decreased as we increased the threshold value from 100% to 33% using the GFD-based algorithm, while using the Haar-based algorithm, the precision maintains a high level of precision accuracy ranging from 94.1% to 100%.

### 4.3. Energy Consumption Efficiency Analysis

The presented distributed processing scheme starts with the cluster-establishing phase where the camera selects a set of nodes to participate in the image processing tasks. The scheme has been decomposed into a set of atomic sub-tasks illustrated in Table 5. We quantified the energy consumption level for each sub-task due to the in-node processing and communication using the Avrora simulator, a tool that emulates the internal resources and processing of a set of sensor nodes such as Mica and TelosB sensors. For instance, the energy consumption of a centralized Haar-based algorithm implementation, where the camera node executes all image processing sub-tasks, can reach 4.02 mJ in 64 × 64-8 bpp images and 4.86 mJ in the 128 × 128-8 bpp image (see Table 5).

**Table 5.** Centralized Haar-based energy consumption for each sub-task.

| Sub-Task | 64 × 64 Pixels 8 bpp | | 128 × 128 Pixels 8 bpp | |
|---|---|---|---|---|
| | Time (s) | Energy (%) | Time (s) | Energy (%) |
| Image decomposition using wavelet decomposition | 0.037 | 0.84 | 0.074 | 1.64 |
| Object extraction | 0.07 | 1.825 | 0.07 | 1.825 |
| Transform ring projection (TRP) | 0.058 | 1.33 | 0.058 | 1.33 |
| Matching using ED | 0.0112 | 0.0256 | 0.0112 | 0.0256 |
| Total | 0.277 | 4.02 | 0.213 | 4.86 |

On the other hand, distributing the image processing tasks across more than one node enables the camera node to initiate more sensing cycles, eventually extending the network lifetime.

A typical distribution would assign a sub-task to an individual node. Thus, a cluster would consist of the camera and three cooperating nodes. However, empirical results have shown that distributing the sub-tasks among two cooperating nodes is the optimal energy-saving choice because it avoids excess communication overhead and unfair processing load distribution.

We investigate the related energy consumption and elapsed time for cluster establishment where the camera selects possible candidate processing nodes, as shown in Table 6. Note that we neglect the energy consumed to capture the image by the camera node.

**Table 6.** Energy consumption and time estimation for cluster-forming phase.

| Cluster-Forming Task | Camera | | Single Neighbor Node | |
|---|---|---|---|---|
| | Time (s) | Energy (%) | Time (s) | Energy (%) |
| Neighborhood Energy Request | 0.000025 | 0.005 | 0 | 0.006 |
| Neighbor Energy Responses | 0 | 0.055 | 0.000025 | 0.005 |
| Cluster Forming and Acknowledgment | 0.00005 | 0.12 | 0.000025 | 0.011 |
| Total | 0.015 | 0.18 | 0.00005 | 0.021 |

After the camera forms the processing cluster and the selected co-nodes acknowledge their participation roles, the camera and P1 cooperate to further object identification and feature extraction. The energy consumption is summarized for the image sizes of 64 × 64 8 bpp and 128 × 128 8 bpp in Tables 7 and 8 for Haar-based and GFD-based distributed schemes, respectively.

**Table 7.** Energy and time estimation for object identification and feature extraction using Haar wavelet transformation.

| | 64 × 64 Pixels 8 bpp | | | | 128 × 128 Pixels 8 bpp | | | |
|---|---|---|---|---|---|---|---|---|
| | Camera | | P1 | | Camera | | P1 | |
| | Time (s) | Energy (%) | Time (s) | Energy (%) | Time (s) | Energy (%) | Time (s) | Energy (%) |
| Apply Haar wavelet transformation | 0.037 | 0.84 | - | - | 0.074 | 1.64 | _ | _ |
| Camera sends low-band coefficients to P1 | 0.000025 | 0.4 | 0 | 0.45 | 0.000025 | 0.4 | 0 | 0.45 |
| P1 extracts and normalizes the object | - | - | 0.131 | 2.94 | - | - | 0.131 | 2.94 |
| Total | 0.037025 | 1.24 | 0.131 | 3.39 | 0.074025 | 2.04 | 0.131 | 3.39 |

**Table 8.** Energy and time estimation for object identification and feature extraction using GFD transformation.

| Sub-Tasks | 64 × 64 Pixels 8 bpp | | | | 128 × 128 Pixels 8 bpp | | | |
| | Camera | | P1 | | Camera | | P1 | |
| | Time (s) | Energy (%) | Time (s) | Energy (%) | Time (s) | Energy (%) | Time (s) | Energy (%) |
|---|---|---|---|---|---|---|---|---|
| Camera extracts ROI | 0.118 | 2.25 | - | - | 0.39 | 9.06 | - | - |
| Camera sends ROI to P1 | 0.000025 | 0.134 | 0 | 0.146 | 0.000025 | 0.453 | 0 | 0.496 |
| P1 extracts the GFD vectors | | | 0.131 | 2.94 | | | 0.131 | 2.94 |
| Total | 0.118025 | 2.384 | 0.131 | 3.086 | 0.390025 | 9.513 | 0.131 | 3.436 |

When the first co-node (P1) receives the region ROI from the camera, it will start the extraction of the object descriptors from the background and normalize the object to be in the center of the image for the Haar-based scheme, while this step is not necessary for GFD due to its invariant properties for object orientation as illustrated in Section 3. Then, the first co-node (P1) will send the normalized approximate coefficients to the second co-node (P2), which in turn will apply the transformation of ring projection (TRP). This process will convert the 2D approximate coefficients to 1D feature vectors for the matching process. Tables 9 and 10 illustrate the processing load's balanced distribution among P1 and P2 for both distributed schemes in detail.

**Table 9.** Energy and time estimation for matching the extracted Haar-based feature vectors.

| Sub-Tasks | P1 | | P2 | |
| | Time (s) | Energy (%) | Time (s) | Energy (%) |
|---|---|---|---|---|
| P1 normalizes object | 0.07 | 1.825 | – | – |
| P1 sends coefficients to P2 | 0.000025 | 0.05121024 | 0 | 0.056 |
| P2 transforms 2D vectors using ring projection | – | – | 0.058 | 1.33 |
| P2 matches the vectors to a reference | – | – | 0.0112 | 0.0256 |
| Total | 0.070025 | 1.87621 | 0.0692 | 1.4116 |

**Table 10.** Energy and time estimation for matching the extracted GFD-based feature vectors.

| Sub-Tasks | P1 | | P2 | |
| | Time (s) | Energy (%) | Time (s) | Energy (%) |
|---|---|---|---|---|
| P1 sends GFD vectors to P2 | 0.000025 | 0.056 | 0 | 0.062 |
| P2 matches the vectors to a reference | – | – | 0.21 | 4.8 |
| Total | 0.000025 | 0.056 | 0.21 | 4.862 |

When the target is recognized, the scheme will notify the end-user with different possible notification message types. Figure 6 shows the energy consumed when a camera node sends a simple 1-byte notification to the end-user. In addition, we show the energy consumption when the extracted feature vectors are added to the notification packets sent to the end-user. Using the feature set extracted with the Haar-based scheme as a notification option will decrease the energy needed for transmission compared to the extracted set based on GFD.

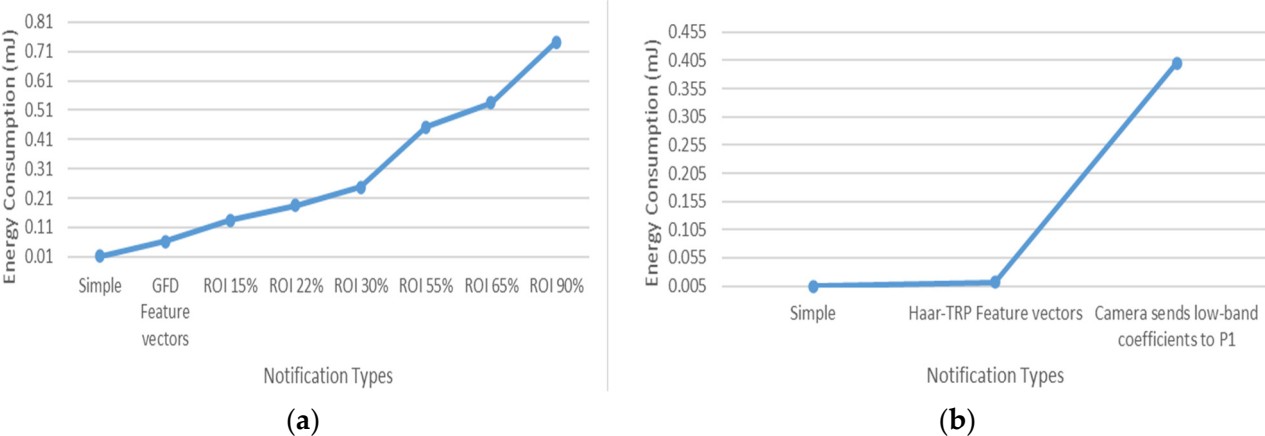

**Figure 6.** Energy consumption for notification in (**a**) scheme based on GFD and (**b**) scheme based on Haar wavelet transform.

Figure 7 summarizes the total energy consumption and elapsed time between the camera and the cooperating processing nodes P1 and P2. As shown, the presented energy consumption level based on the Haar wavelet shows a more balanced processing load distribution using the 64 × 64 8 bpp and 128 × 128 8 bpp images than that using GFD. The energy consumption related to the processing of the scheme is shared between the different nodes, where the camera node consumes around 27% of energy during a single sensing cycle, while the cooperative nodes consume 73% of energy from the total consumed energy required to process the scheme using an image size of 64 by 64. However, for an image of 128 × 128 8 bpp, these percentages of energy consumption could reach 37% in the camera node and 63% in collaborating nodes. The camera elects candidate nodes for cluster participation based on the highest residual energy in each new sensing cycle. This step will distribute the total collaborated energy consumption over the alive nodes, consequently extending the node lifetime.

Figure 8 shows our presented scheme's energy consumption and elapsed time compared to the use of the general Fourier descriptors (GFDs) for feature extraction. As we infer from Figure 8a,b, using Haar wavelet decomposition, the amount of energy consumed by P1 and P2 is the same regardless of the size of the original image. The Haar decomposition result will permanently be fixed to the size 64 × 64 8 bpp. We also note that applying the Haar wavelet to extract features considerably preserves the camera energy in the case of larger images 128 × 128 bpp. The node P2 is responsible for measuring the distance between the extracted signature and the reference signature. The length of the feature vector implies the complexity of this matching. Therefore, we note that P2 consumed more energy when using the GFD (52) feature vector than when using the Haar feature vector (12). There is also noteworthy energy preserving in the node (P1) as the complexity of the TRP process is less demanding than the GFD process. Figure 8c,d show that Haar wavelet decomposition decreases the needed time for a single cycle compared to the GFD method. It is also evident that the time required by the Haar-based method is fixed regardless of the image size.

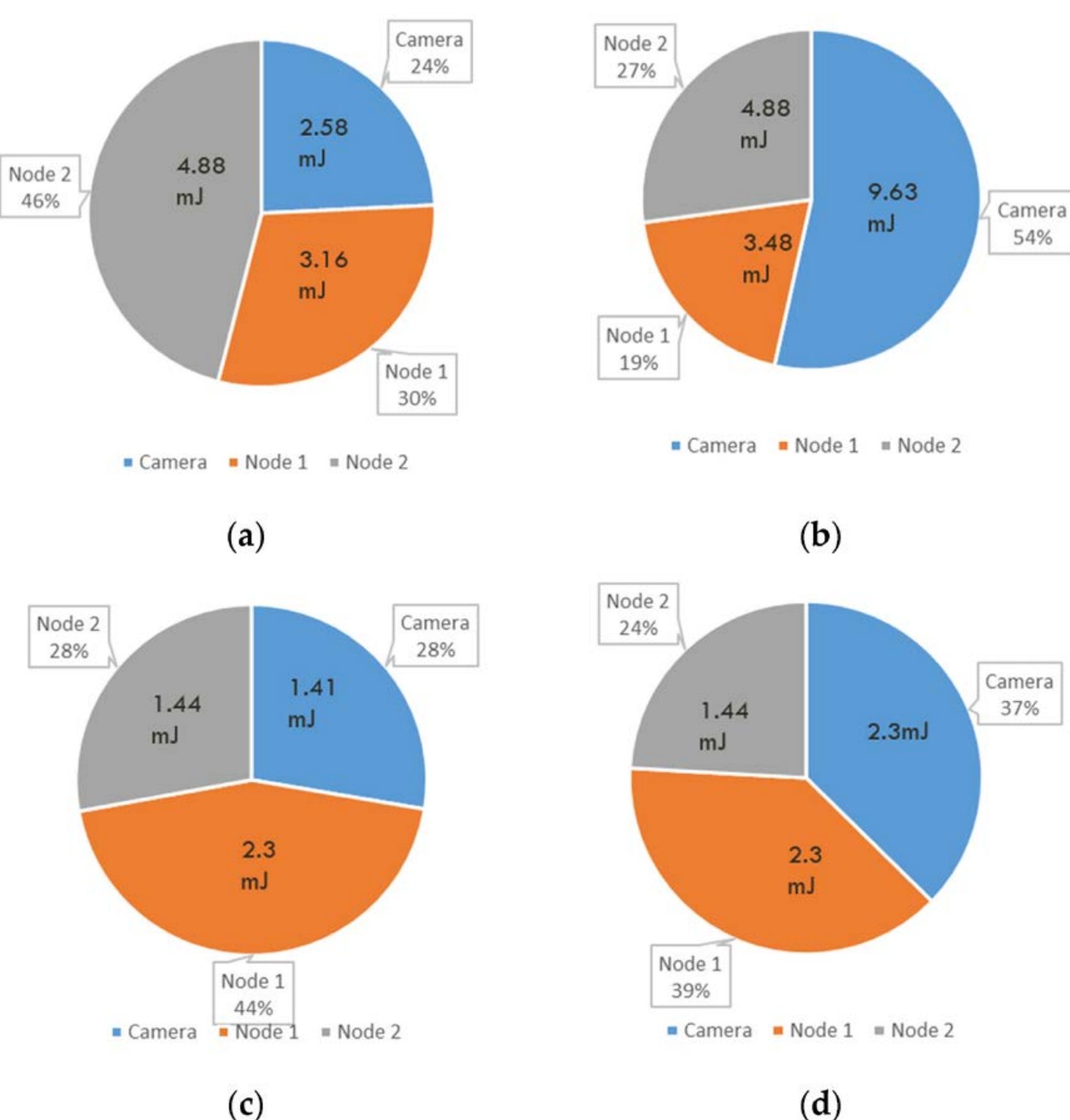

**Figure 7.** Comparing total per-node time and energy consumption in the processing cluster: (**a**) scheme based on GFD using image of size 64 by 64-8 bpps; (**b**) scheme based on GFD using image of size 128 by 128-8 bpps; (**c**) scheme based on Haar wavelet transform using image of size 64 by 64-8 bpps; (**d**) scheme based on Haar wavelet transform using image of size 128 by 128-8 bpps.

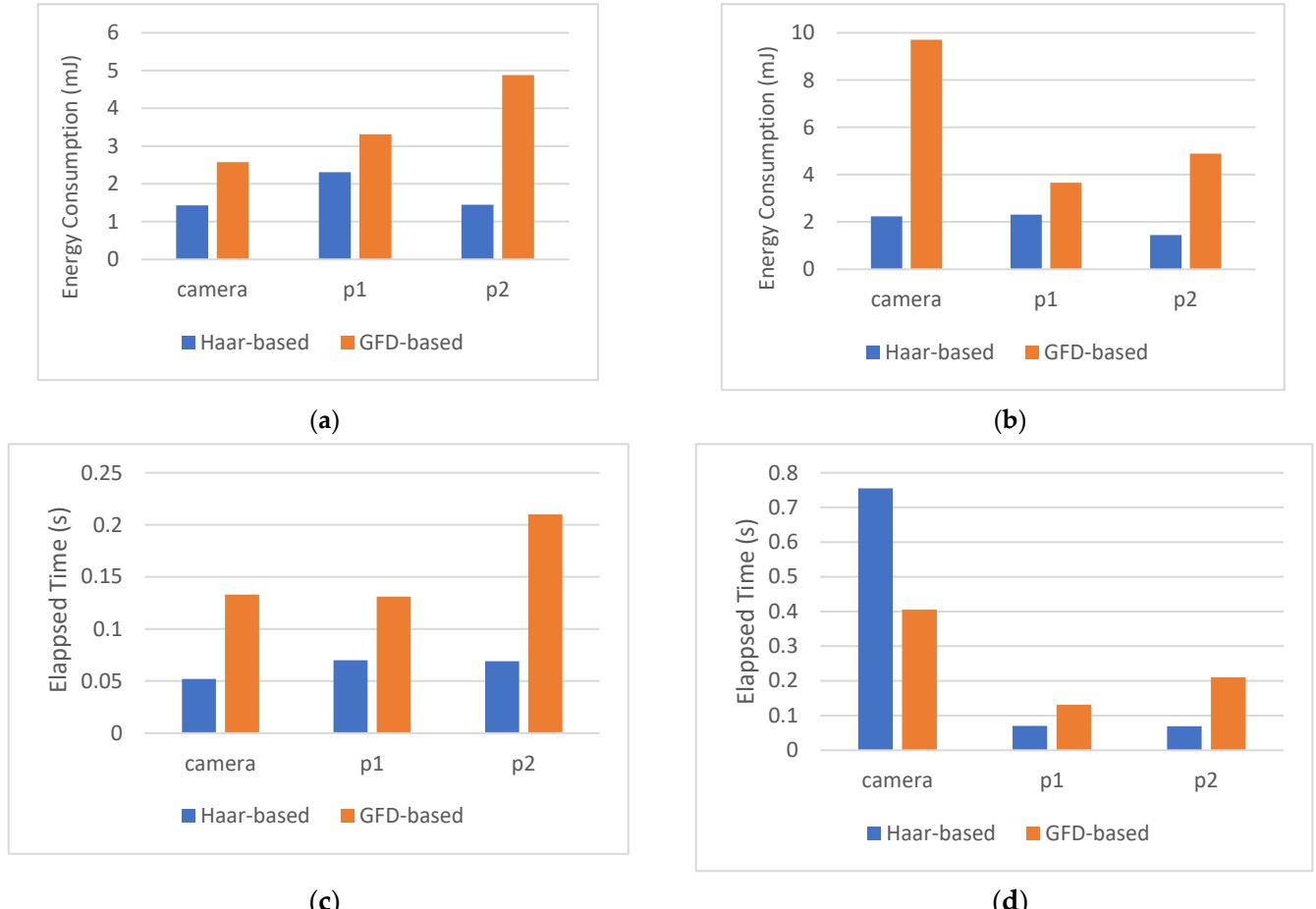

**Figure 8.** Energy consumption and elapsed time per sensing cycle. (**a**) Energy consumption per sensing cycle, image $64 \times 64$-8 bpp; (**b**) energy consumption per sensing cycle, image $128 \times 128$-8 bpp; (**c**) elapsed time per sensing cycle, image $64 \times 64$ bpp; (**d**) elapsed time per sensing cycle, image $128 \times 128$ bpp.

Figure 9 plots the energy consumption level in the first five sensing cycles to study the energy level distribution between the camera and the ten cooperating nodes. In each sensing cycle, the camera selects two nodes with the highest residual energy to ensure that participating in the processing cluster will prolong the network lifetime as much as possible. The presented cumulative energy consumption level shows that each node is selected once during every five sensing cycles. The figure shows that the camera repeatedly participates in every sensing cycle while other nodes are alternately selected based on their residual energy level. This is due to the leading role that the camera plays in leading the processing cluster. Therefore, we note that the camera energy is depleted before the energy of any other processing nodes. The same behavior is observed in image size $128 \times 128$ 8 bpps, as shown in Figure 9b.

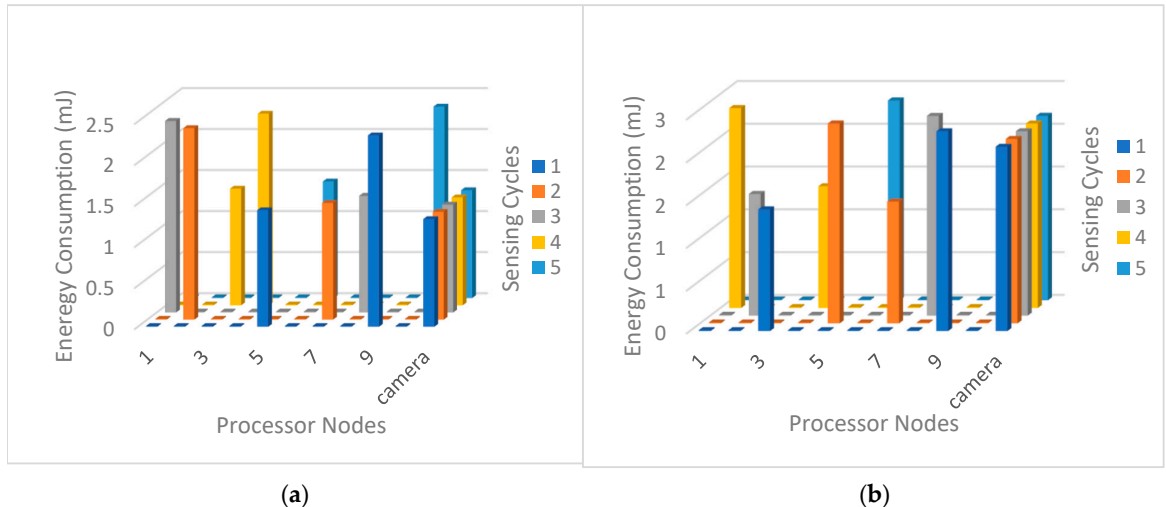

**Figure 9.** Distribution of energy consumption load per sensing cycle for the first five sensing cycles using (**a**) images with a size of $64 \times 64 \times 8$ bpp and (**b**) images with a size of $128 \times 128 \times 8$ bpp.

Figure 10 shows the centralized implementation of the Haar wavelet scheme, where the camera is responsible for processing all recognition tasks. The graph plots the cumulative energy consumption in the camera from the first sensing cycle until the camera depletes its energy. We note that the centralized scheme can reach only 20 sensing cycles.

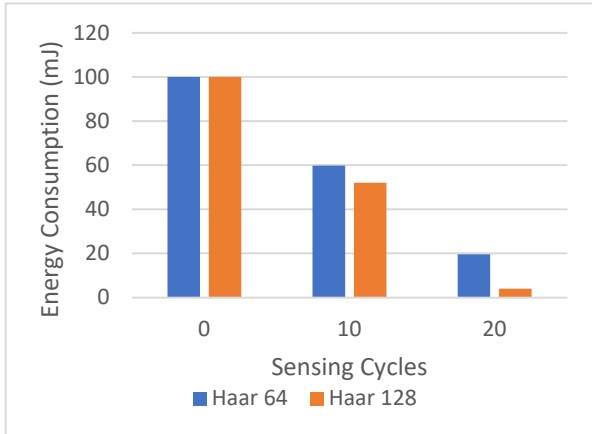

**Figure 10.** Energy consumption in camera, centralized Haar-based recognition scheme.

On the other hand, the distributed implementation of Haar-based image recognition can extend the camera lifetime, as demonstrated in Figure 11. As sensing cycles proceed, each node will have different residual energy based on its participation role in the processing cluster. Figure 11 gradually plots the per-node energy consumption (estimated average) every ten sensing cycles. In the distributed implementation of the Haar-based recognition scheme, the sensing cycles extended from 20 sensing cycles in centralizing to 70. Note that when the image size is $128 \times 128$ 8 bpp, the number of sensing cycles is decreased from 70 sensing cycles, as shown in Figure 11a, to 40 sensing cycles, as shown in Figure 11b.

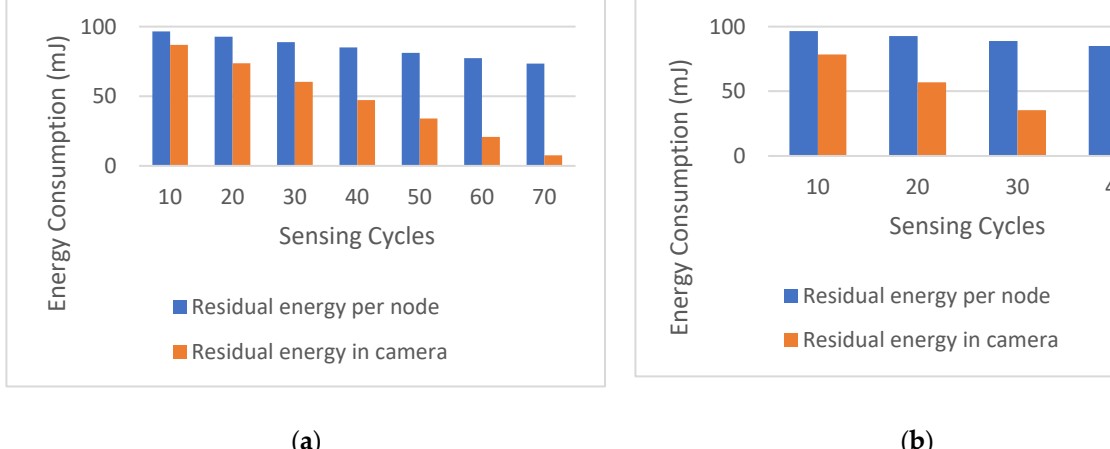

(**a**)                                          (**b**)

**Figure 11.** Cumulative residual energy in the nodes in Haar-based scheme for (**a**) 64 × 64 × 8 bpp images and (**b**) 128 × 128 × 8 bpp images.

This decrease can be attributed to the increased energy consumption in the camera node when it decomposes the larger image size (128 × 128 bpp) to extract the Haar coefficients. As the sensing cycles iterate, the amount of energy consumed increases. The amount of energy consumed does not exceed 30% of per-node residual energy in both image sizes. This implies that the scheme preserves approximately at least 70% of the per-node residual energy level, which is a promising indicator that adopting the proposed scheme can be applied efficiently within a multi-application network that can reuse the nodes for other sensing purposes.

Figure 12 shows the energy consumption of a distributed scheme based on GFD. The figure plots the average camera residual energy in contrast to the average residual energy in network nodes.

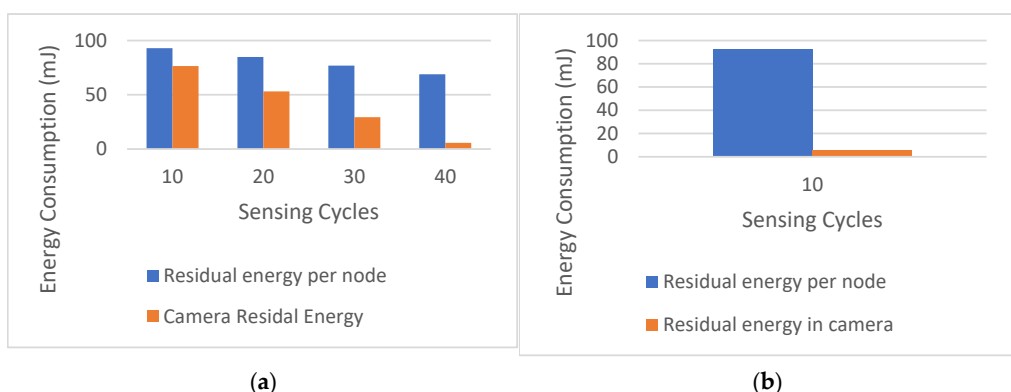

(**a**)                                          (**b**)

**Figure 12.** Cumulative residual energy in the nodes in the GFD-based scheme for (**a**) 64 × 64 × 8 bpp images and (**b**) 128 × 128 × 8 bpp images.

We can see that the proposed scheme prolongs the camera lifetime to accomplish 70 sensing cycles instead of 40 cycles using image size 64 × 64 8 bpp as presented in Figures 11a and 12a where it completes 40 sensing cycles instead of 10 sensing cycles using GFD on image size 128 × 128 bpp, as presented in Figures 11b and 12b.

Moreover, we can infer that using the Haar wavelet also extends the average residual energy level in network nodes where we have at least 70% of the energy level where it is the same residual energy level using the GFD.

As illustrated in Figure 13, the elapsed time for a multiple sensing cycle remarkably decreased to 2 s, whereas using GFD needs between 5 and 8 s depending on image size. As shown, we can simultaneously perform 20 sensing cycles using the Haar-based scheme to accomplish only 10 sensing cycles using the GFD-based scheme on image size

64 × 64 8 bpp and 40 sensing cycles using the presented scheme to accomplish only 10 sensing cycles using GFD on image size 128 × 128 8 bpp.

**Figure 13.** Average elapsed time per 10th sensing cycle in network.

This is evidence that minimizing the number of candidate collaborating nodes raises the distributed processing loads on the network nodes. Nevertheless, using only two collaborating nodes will rapidly drain the network node energy level due to maximizing each node's selection probability. Conversely, assigning tasks to other network nodes helps the camera process more sensing cycles, extending its lifetime.

Previous work in the literature has investigated various approaches to energy efficiency in event-based multimedia sensing. We can roughly classify previous work, as shown in Table 11, across two dimensions: the processing model and the implementation approach. Similar to our work, some approaches distributed the work across more than one node, while others adopted a centralized approach where a single node executes all the work. It is also noted that some approaches were implemented using hardware components instead of a software solution.

**Table 11.** Comparison with related works.

| Related Work | Processing Model | Schema Based On | Implementation Approach | Per-Node Energy Consumption in Comparison to the Presented Solution |
|---|---|---|---|---|
| [30] | Local | Local event-based detection based on centroid distance and histogram algorithms | Software | Higher energy consumption |
| [31] | Local | Local event-based detection based on GFD | Software | Higher energy consumption |
| [5] | Distributed | Curve fitting technique | Software | Higher energy consumption |
| [19] | Local | Discrete Tchebichef transform (DTT) | Software | Higher energy consumption |
| [27] | Distributed | Distributed compression | Software | Higher energy consumption |
| [28] | Local | Object extraction scheme | Hardware | Lower energy but with a very high implementation cost |
| [36] | Local | Quad-tree decomposition | Software | Higher energy consumption |
| [25] | Distributed | Face-detection algorithm using discriminative vectors | Software | Higher energy consumption |
| GFD | Distributed | Distributed event-based detection and recognition using GFD | Software | Higher energy consumption |
| Presented Approach | Distributed | Distributed event-based detection and recognition using Haar wavelet | Software | – |

In [30], the authors presented a centralized event-based detection solution using centroid distance and histogram methods. It has been shown that a multimedia node

consumes 47.6 mJ for an image size of $64 \times 64 \times 8$ bpp and 80.2 mJ for an image size of $128 \times 128 \times 8$ bpp. These levels of energy are much higher than the energy required in this proposed new solution. Furthermore, the centroid distance exhibits a low level of accuracy as a descriptor for target recognition [57].

In our previous work published in [31], we compared the energy efficiency of a scheme based on GFD and another one based on Zernike moments (ZMs) for event-based object recognition. We found that the energy consumption using ZMs can reach 9.995 mJ when using images of $64 \times 64$-8 bpp and 16.8 mJ for images of $128 \times 128$-8 bpp, while the required energy in a scheme based on GFD is 4.02 mJ for images of $64 \times 64$-8 bpp and 4.86 mJ for images of $128 \times 128$-8 bpp. Compared to our new solution, based on distributed scheme implementation, the energy consumption in the camera node decreased to 1.4 mJ instead of 2.46 mJ using GFD (approx. 50% reduction) in all image sizes. This energy expansion will extend the camera life and improve the network performance.

In [25], the authors introduced an energy-aware face-detection algorithm that utilizes a lightweight feature vector to be sent to the sink at a low transmission cost. Despite its demonstrated low energy consumption, this technique can flood the network with unnecessary data if the end-user is looking for a specific target. Performing the recognition algorithm in the network rather than on the sink can reduce the bandwidth from irrelevant data and can consequently increase the network lifetime.

Some other research work used the approach of energy-aware image compression to reduce the amount of transmitted data through the network. In [5], the authors have proposed an energy-aware scheme intended for image transmission in WMSNs. Their approach ensures a low overhead data compression for energy saving based on the curve fitting technique. The obtained results demonstrated energy efficiency compared to other similar data compression algorithms, but in a comparison with the proposed distributed scheme, we can note that our approach achieved much better performances in terms of energy consumption.

In [19], Kouadria et al. used a discrete Tchebichef transform (DTT)-based image compression technique. Due to its lower complexity, the DTT compression technique is an alternative to the discrete cosine transform (DCT). However, experimental results have shown that it consumes a considerable amount of energy per block of $8 \times 8$ pixels (around 146.63 mJ, which is a very high level of consumed energy).

In [27], the authors introduced a distributed compression algorithm. It is noted that the approach consumes around 1.4 J for the compression of an image size of $(512 \times 512)$ 8 bpp, which is considered an extremely high level of energy consumption. In the same context, G. Nikolakopoulos et al. presented a compression scheme based on quad-tree decomposition in [36]. The obtained results showed that it consumed 120 mJ energy to transmit an image of $128 \times 128 \times 8$ bpp and 45 mJ to transmit an image of $64 \times 64 \times 8$ bpp.

As a solution to reduce the high energy consumption related to the software implementation of compression algorithms, another approach based on hardware implementation was proposed in [28]. Although the hardware implementation increases the cost, it ensures a significant gain in energy. However, we have shown that we can identify and recognize events of interest without altering the sensor design while maintaining a low energy consumption level.

In conclusion, the new approach of distributed implementation proved that the processing load of the camera sensor was reduced. At the same time, other tasks were transferred to other cooperating nodes, which extends the sustainability of the application and allows the camera node to execute more sensing cycles. The performance evaluation of the presented scheme shows that our work outperforms the other proposed solution in the literature in terms of energy consumption associated with the target recognition in image-based sensing which consequently extends the multimedia application lifetime in the wireless sensor network.



## 5. Conclusions

This paper addressed the specification and design of a low-energy processing scheme intended to be deployed in distributed cluster-based implementation using Haar wavelet decomposition to extract the feature vector from the region of interest (ROI). In this work, we presented our experimental results of the distributed implementation of a proposed target detection scheme based on a multimedia sensor and a wireless sensor network. In this approach, the results have shown that the processing load was decreased in the camera to only 1.4 mJ in the distributed processing cluster using image size $64 \times 64 \times 8$ bpp and 2.46 mJ using image size $128 \times 128 \times 8$ bpp. It was also shown that using the Haar wavelet features prolonged the camera life to accomplish 70 sensing cycles instead of 43 cycles with an image size of $64 \times 64$ 8 bpp and 44 sensing cycles instead of 10 sensing cycles using GFD on an image size of $128 \times 128$ bpp. The energy consumption related to processing is split among the network, approximately 27% in the camera and 73% in cooperative nodes. Based on these results, we believe that the distributed approach can significantly extend the life of the camera node, which testifies to its efficiency for IoT multimedia sensing.

As future work, we think that the proposed scheme needs to be validated using a formal verification approach before its real experimentation on a testbed platform. Furthermore, we also believe that the design of an adequate lightweight security approach is crucial to protect the exchanged data inside of the processing cluster of the IoMT platform and the remote-control server [58].

**Author Contributions:** All the authors participated in the development of the proposed work, in the discussion of the results, and in writing the paper. All authors have read and agreed to the published version of the manuscript.

**Funding:** This research received no external funding.

**Data Availability Statement:** Not applicable.

**Acknowledgments:** This research project was supported by a grant from the "Research Center of College of Computer and Information Sciences", Deanship of Scientific Research, King Saud University.

**Conflicts of Interest:** The authors declare no conflict of interest.

# Appendix A

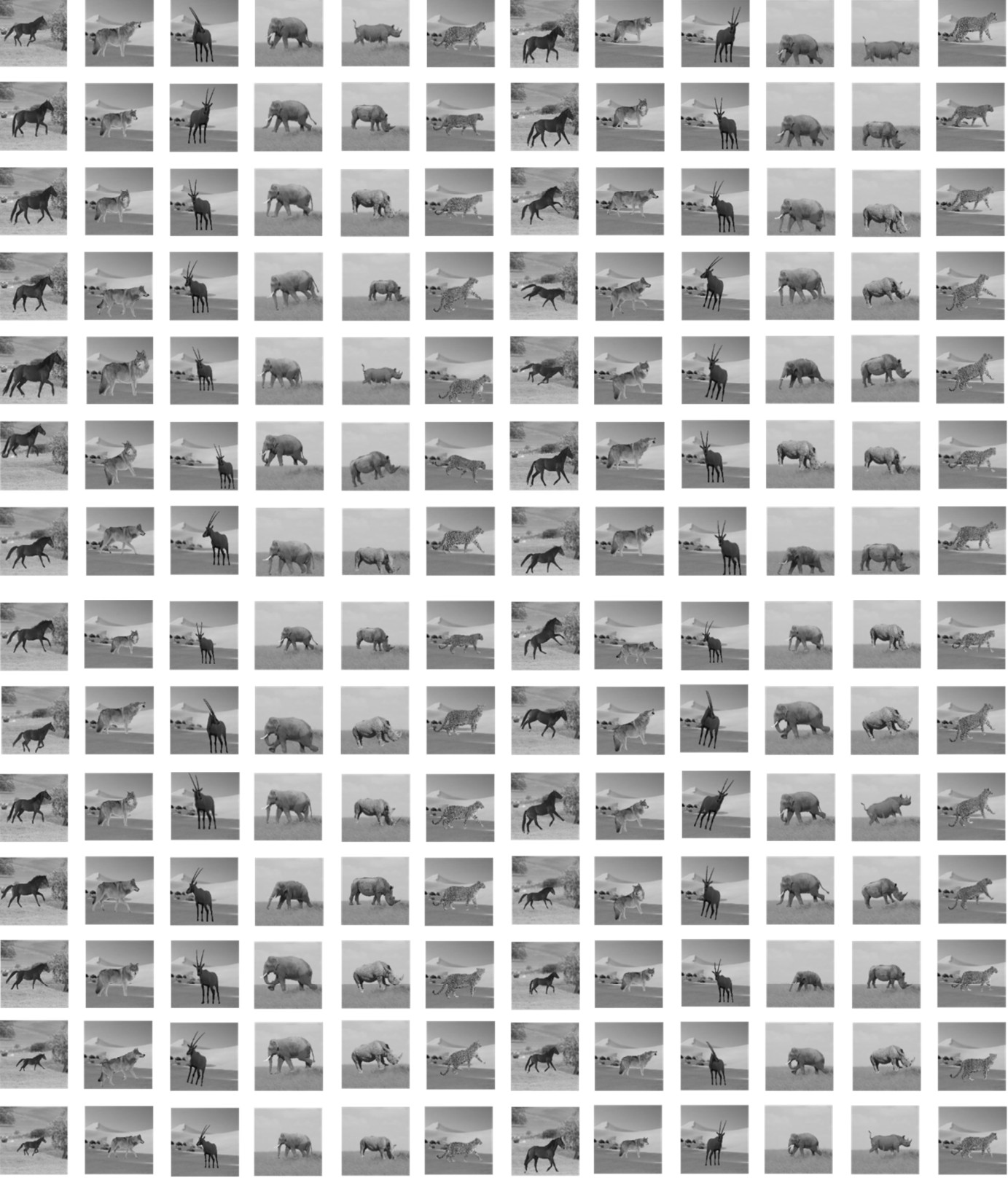

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
