# Peer review of "A Study on Energy Efficiency of a Distributed Processing Scheme for Image-Based Target Recognition for Internet of Multimedia Things"

_computers, doi:10.3390/computers12050099_

Round 1

Reviewer 1 Report

The article objectives are clear and well presented. It presents an approach for distributed event-based sensing achieved by a cluster of processing nodes based on the Haar Wavelet Transform of the image in comparison with the scheme based on a General Fourier shape descriptor in the context of recognition accuracy of the target as well as the energy consumption.

The main contribution of the authors is the proposed distributed approach, which offers a low energy consumption associated with clustering.

The proposed methodology and solution, based on combination of existing and well-known technologies in real environment make the author’s work the state-of-the art and actual.

The work is well structured and English language looks good. The article is written intelligibly in a very good scientific style and it is easy to read. The title and abstract reflect very well to the content. The conclusion very well summarized the proposed work.

The references are relevant to the topic.

The figures are with good quality. However, I offer to the authors to change the type of fig. 8 with 4 or 5 different figures. I believe that it will be more readable.

Author Response

Dear Reviewer , 

Please find attached our response to your comments 

Best regards

Adel Soudani

Reviewer 2 Report

Certainly the Internet of Multimedia Things (IoMT) is booming. I agree that event-based sensing schemes can be used to provide efficient data transmission and longer network lifetime. In this paper, an approach for distributed event-based sensing using a cluster of processing nodes is proposed. The authors trade off the processing load among the nodes in the cluster. The efficiency comparison is interesting, the results look encouraging.

I have several suggestions regarding the article. First, I would recommend a revision of the language as there are certain expressions that make it difficult to read.

I am struck by the fact that security is not mentioned among the most relevant problems of IoMT and IoT. In works such as "A Test Environment for Wireless Hacking in Domestic IoT Scenarios" there are security problems shared by both scenarios. On the other hand, the proposed scenario presents a protocol but I miss some formal method for validation as "AVISPA in the validation of Ambient Intelligence Scenarios".

Certainly the Internet of Multimedia Things (IoMT) is booming. I agree that event-based sensing schemes can be used to provide efficient data transmission and longer network lifetime. In this paper, an approach for distributed event-based sensing using a cluster of processing nodes is proposed. The authors trade off the processing load among the nodes in the cluster. The efficiency comparison is interesting, the results look encouraging.

I have several suggestions regarding the article. First, I would recommend a revision of the language as there are certain expressions that make it difficult to read.

I am struck by the fact that security is not mentioned among the most relevant problems of IoMT and IoT. In works such as "A Test Environment for Wireless Hacking in Domestic IoT Scenarios" there are security problems shared by both scenarios. On the other hand, the proposed scenario presents a protocol but I miss some formal method for validation as "AVISPA in the validation of Ambient Intelligence Scenarios".

Author Response

Dear Reviewer , 

Please find attached our response to your comments 

Best regards

Reviewer 3 Report

Paper formatting can be improved. Tables and figures can be rearranged for better typesetting.

English is fluent, authors just have to fix some typos.